# Levels of SARS-CoV-2 population exposure are considerably higher than suggested by seroprevalence surveys

**Siyu Chen**[1], **Jennifer A. Flegg**[2], **Lisa J. White**[1], **Ricardo Aguas**[3]*

**1** Big Data Institute, Li Ka Shing Centre for Health Information and Discovery, Nuffield Department of Medicine, University of Oxford, United Kingdom, **2** School of Mathematics and Statistics, University of Melbourne, Melbourne, Australia, **3** Centre for Tropical Medicine and Global Health, Nuffield Department of Medicine, University of Oxford, Oxford, United Kingdom

* ricardo@tropmedres.ac

## Abstract

Accurate knowledge of prior population exposure has critical ramifications for preparedness plans for future SARS-CoV-2 epidemic waves and vaccine prioritization strategies. Serological studies can be used to estimate levels of past exposure and thus position populations in their epidemic timeline. To circumvent biases introduced by the decay in antibody titers over time, methods for estimating population exposure should account for seroreversion, to reflect that changes in seroprevalence measures over time are the net effect of increases due to recent transmission and decreases due to antibody waning. Here, we present a new method that combines multiple datasets (serology, mortality, and virus positivity ratios) to estimate seroreversion time and infection fatality ratios (IFR) and simultaneously infer population exposure levels. The results indicate that the average time to seroreversion is around six months, IFR is 0.54% to 1.3%, and true exposure may be more than double the current seroprevalence levels reported for several regions of England.

## Author summary

It is particularly challenging to determine the true proportion of the population that has been previously exposed to SARS-CoV-2. Serological surveys that measure how many people have antibodies against the virus are a promising tool but results from such studies need to be interpreted carefully. Several studies following individuals over time after they've had a known infection were able to determine that antibodies are only measurable up to 6–9 months, on average. The immediate implication is that serological studies will inevitably under-estimate the number of people exposed, since some will have a lower antibody count when the study is conducted and test negative. We propose a method that takes this into account and informs the true level of exposure from triangulating serological data with mortality and test positivity data.

**Data Availability Statement:** All data, code, and materials used in the analyses can be accessed at: https://github.com/SiyuChenOxf/ COVID19SeroModel/tree/master. All parameter

estimates and figures presented can be reproduced using the codes provided.

**Funding:** JF acknowledges funding from the Australian Research Council (DP200100747). LJW is funded by the Li Ka Shing Foundation and the University of Oxford's COVID-19 Research Response Fund (BRD00230). RA is funded by the Bill and Melinda Gates Foundation (OPP1193472). The funders had no role in study design, data collection and analysis, decision to publish, or preparation of the manuscript.

**Competing interests:** The authors have declared that no competing interests exist.

## Introduction

The COVID-19 pandemic has inflicted devastating effects on global populations and economies [1]. Levels and styles of reporting epidemic progress vary considerably across countries [2], with cases consistently being under-reported and case definitions changing considerably over time. Therefore, the scientific and public health communities turned to serological surveys as a means to position populations along their expected epidemic timeline and thus provide valuable insights into COVID-19 lethality [3,4]. Those prospects were frustrated by apparent rapid declines in antibody levels following infection [5–7]. Population-wide antibody prevalence measurements can significantly underestimate the level of underlying population immunity, with obvious implications for intervention strategy design and vaccine impact measurement.

Continued research efforts to determine the correlates for protective immunity against disease and infection have found that while antibody titers are poor indicators of sustained immunity, cellular immunity can play a determinant role in limiting susceptibility to further SARS-CoV-2 challenges in previously exposed individuals [8,9]. Unfortunately, performing T cell assays at scale is technically challenging and expensive, which justified the decision to conduct a series of serology surveys (some of which are still underway) in many locations globally to provide a better understanding of the extent of viral spread among populations [10].

In England, a nationwide survey sampling more than 100,000 adults was performed from 20 June to 13 July 2020. The results suggested that 13% and 6% of the population of London and England, respectively, had been exposed to SARS-CoV-2, giving an estimated overall infection fatality ratio (IFR) of 0.90% [11]. Although corrections were made for the sensitivity and specificity of the test used to infer seroprevalence, declining antibody levels were not accounted for. This is a limitation of the approach, potentially resulting in underestimates of the true levels of population exposure [12] and an overestimate of the IFR.

We now have a much clearer picture of the time dynamics of humoral responses following SARS-CoV-2 exposure, with antibody titers remaining detectable for approximately 6 months [13,14]. Commonly used serological assays have a limit of antibody titer detection below which a negative result is yielded. Hence, a negative result does not necessarily imply an absence of antibodies, but rather that there is a dynamic process by which the production of antigen-targeted antibodies diminishes once infection has been resolved, resulting in decaying antibody titers over time. As antibody levels decrease below the limit of detection, seroreversion occurs.

We define the seroreversion rate as the inverse of the average time taken following seroconversion for antibody levels to decline below the cut-off point for testing seropositive. In a longitudinal follow-up study, antibodies remained detectable for at least 100 days [6]. In another study [15], seroprevalence declined by 26% in approximately three months, which translates to an average time to seroreversion of around 200 days. However, this was not a cohort study, so newly admitted individuals could have seroconverted while others transitioned from positive to negative between rounds, leading to an overestimation of the time to seroreversion.

Intuitively, if serology were a true measure of past exposure, we would expect a continually increasing prevalence of seropositive individuals over time. However, data suggest this is not the case [16], with most regions in England showing a peak in seroprevalence at the end of May 2020. This suggests seroreversion plays a significant role in shaping the seroprevalence curves in England and that the time since the first epidemic peak will influence the extent to which subsequent seroprevalence measurements underestimate the underlying population attack size (proportion of the population exposed). We argue that the number of people infected over the course of the epidemic can be informed by data triangulation, i.e., by

**Fig 1. Progression of exposed individuals through the various clinical (below the timeline) and diagnostic (above the timeline) stages of infection and recovery.** Stages marked in grey represent events that may happen, with a probability consistent with the darkness of the shade of grey.

combining numbers of deceased and seropositive individuals over time. For this linkage to be meaningful, we need to carefully consider the typical SARS-CoV-2 infection and recovery timeline (Fig 1).

Most individuals, once infected, experience an incubation period of approximately 4.8 days (95% confidence interval (CI): 4.5–5.8) [17], followed by the development of symptoms, which include fever, dry cough, and fatigue, although some individuals will remain asymptomatic throughout. Symptomatic individuals may receive a diagnostic PCR test at any time after symptom onset; the time lag between symptom onset and date of test varies by country and area, depending on local policies and testing capacity. Some individuals might, as their illness progresses, require hospitalization, oxygen therapy, or even intensive care, eventually either dying or recovering.

The day of symptom onset, as the first manifestation of infection, is a critical point for identifying when specific events occur relative to each other along the infection timeline. The mean time from symptom onset to death is estimated to be 17.8 days (95% credible interval (CrI): 16.9–19.2 days) and to hospital discharge 24.7 days (22.9–28.1 days) [18]. The median seroconversion time for IgG (long-lasting antibodies thought to be indicators of prior exposure) is estimated to be 14 days post-symptom onset; the presence of antibodies is detectable in less than 40% of patients within 1 week of symptom onset, rapidly increasing to 79.8% (IgG) at day 15 post-onset [19]. We assume that onset of symptoms occurs at day 5 post-infection and that it takes an average of 2 additional days for people to have a PCR test. Thus, we fix the time lag between exposure and seroconversion, $\delta_\epsilon$, at 21 days; the time lag between a PCR test and death, $\delta_P$, at 14 days; and assume that seroconversion in individuals who survive occurs at approximately the same time as death for those who do not (Fig 1).

Thus, we propose to use population-level dynamics (changes in mortality and seroprevalence over time) to estimate three key quantities: the seroreversion rate, the IFR, and the total population exposure over time. We developed a Bayesian inference method to estimate said quantities, based on official epidemiological reports and a time series of serology data from blood donors in England, stratified by region [16] (see Materials and Methods for more details). This dataset informed the national COVID-19 serological surveillance, and its data collection was synchronous with the 'REACT' study [11]. The two serosurveys use different, but comparable, antibody diagnostic tests [20]. While the 'REACT' study used the FORTRESS lateral flow immunoassay (LFIA) test for IgG [11], the data analyzed here were generated using the Euroimmun ELISA (IgG) assay [21]. The independent 'REACT' study acts as a validation dataset, lending credence to the seroprevalence values used. For example, seroprevalence in London was reported by 'REACT' to be 13.0% (95% CI: 12.3–13.6%)[11] for the period 20 June to 13 July 2020. In comparison, the London blood-donor time series indicated seroprevalence to be 13.3% (95% CI: 8.4–16%) [21] on 21 June 2020. Notably, the Abbott IgG antibody testing assay showed the most striking decline in sensitivity over time compared with

other serological assays [22,23], which limits comparisons across the two datasets as time progresses.

We developed a method that combines daily mortality data with seroprevalence in England, using a mechanistic mathematical model to infer the temporal trends of exposure and seroprevalence during the COVID-19 epidemic. We fit the mathematical model jointly to serological survey data from seven regions in England (London, North West, North East (North East and Yorkshire and the Humber regions), South East, South West, Midlands (East and West Midlands combined), and East of England) using a statistical observation model (see the Materials and Methods section for more details on the input data sources, mechanistic model, and fitting procedure). We considered that mortality is perfectly reported and proceeded to use this anchoring variable to extrapolate the number of people infected 3-weeks prior. We achieved this by estimating region-specific IFRs (defined as $\gamma_i$), which we initially assumed to be time invariant, later relaxing this assumption. The identifiability of the IFR metric was guaranteed by using the serological data described above as a second source of information on exposure. From the moment of exposure, individuals seroconvert a fixed 21 days later and can then serorevert at a rate, $\beta$, that is estimated as a global parameter. We thus have both mortality and seropositivity prevalence informing SARS-CoV-2 exposure over time. Extending from the baseline model thus described, we conducted sensitivity analyses on key assumptions to evaluate the robustness of the results presented in the main paper. These sensitivity analyses explore how estimates for IFRs and seroreversion rates depend on assumptions around the timelines of infection/testing and the data sources used (see the sensitivity analysis subsection in the Materials and Methods for more details).

Several other research groups have used mortality data to extrapolate exposure and as a result provide estimates for IFR. Some IFR estimates have been published assuming serology cross-sectional prevalence to be a true reflection of population exposure, while others used infection numbers generated by mechanistic dynamic models fit to mortality data [24]. Most recently, sophisticated statistical techniques, which take into account the time lag between exposure and seroconversion, have been used to estimate the underlying population exposure from seroprevalence measurements [25], with some also considering seroreversion [26–28]. Our method is very much aligned with the latter studies but is applied at a regional level while using a dataset that has been validated by an independent, largely synchronous study [21] and uses test positivity data to help inform time-varying transmission intensity.

## Results

Results from the fixed IFR inference method show excellent agreement with serological data (Fig 2). We found that, after seroconverting, infected individuals remain seropositive for about 176 days on average (95% CrI: 159–197 days) (Tables 1 and S1, S1 Fig). This relatively rapid (approximately six months) seroreversion is similar to estimates from experimental studies [13,14,23,29,30], and the choice of an exponential distribution for seroreversion seems to be validated by long follow-up longitudinal studies showing antibody persistence up to 1 year [23,30,31], with 59% (95% CrI: 50–68%) of seropositive individuals seroreverting after 52 weeks (S2 Fig). These seroreversion rates are also broadly consistent with the observation of 83% protection against reinfection within 6 months of disease in UK patients [32].

As a consequence of this rapid seroreversion, epidemic progression will result in an increasing gap between measured serology prevalence levels and cumulative population exposure to the virus. Ultimately, this may mean that more than twice as many people have been exposed to the virus relative to the number of people who are seropositive (Fig 2), highlighting the importance of our method in aiding interpretation of serological survey results and their use

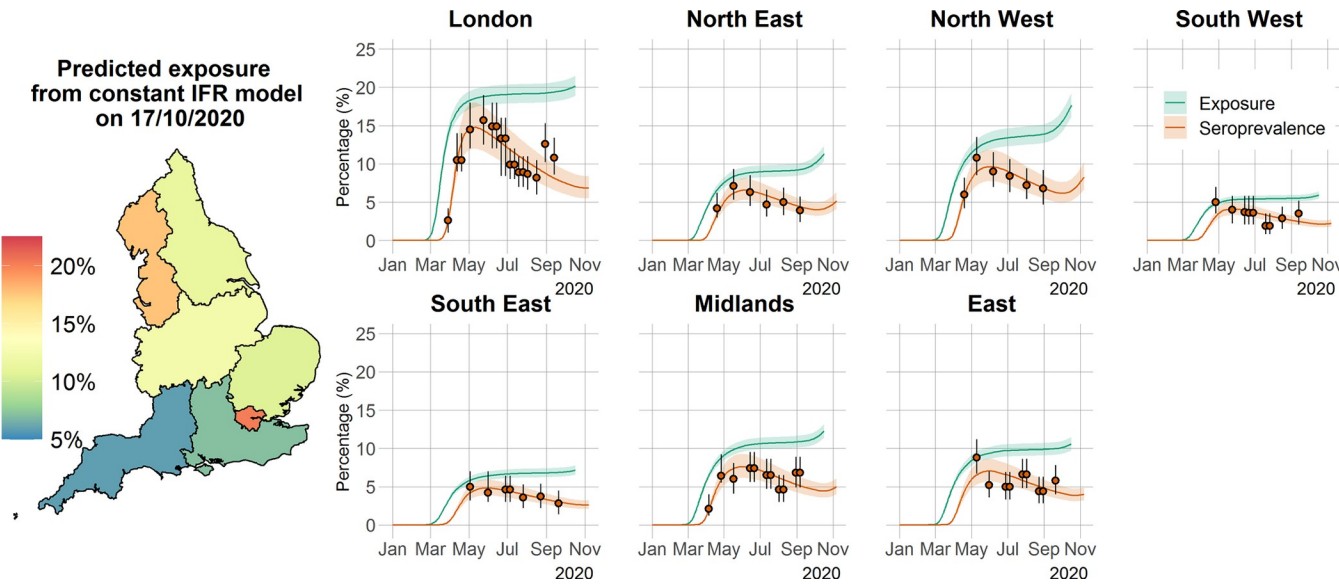

**Fig 2. Time course of the SARS-CoV-2 pandemic up to 7 November 2020 for the seven regions in England in the constant IFR model.** The solid orange circles and black error bars in each regional panel represent the observed seroprevalence data and their credible intervals, respectively, after adjusting for the sensitivity and specificity of the antibody test. The green and orange lines show the model predictions of median exposure and seroprevalence, respectively, while the shaded areas correspond to the 95% CrI. The regional predicted exposure levels (expressed as the proportion of the population that has been infected) as of 17 October 2020 are shown on the map of England. The map base layer was obtained from: https://opendata.arcgis.com/datasets/8d3a9e6e7bd445e2bdcc26cdf007eac7_4.geojson.

for informing policy decisions moving forward. Seroreversion is responsible for decreased seropositivity over periods of continued transmission (as evidenced by mortality and case data) and thus why we had to resort to mortality data to inform the true exposure of populations to SARS-CoV-2. This is made clear by comparing the shapes of the regional cumulative death curves (S3 Fig) with those of the estimated cumulative total exposure (Fig 2).

We also estimated age-independent IFRs for the seven English regions (means ranging from 0.49% to 1.18%; Table 1) that are in very good agreement with other estimates for England [33]. The estimated IFRs were noticeably lower for London and higher for the North East, South East and South West, indicating a clear signal for a lower probability of death per infection in London. Given there are no significant disparities in treatment outcomes across regions [34], we explored several demographic and epidemiological factors that could explain the observed trend (S4 Fig). There is a strong positive correlation between the proportion of the population over the age of 45 years (when disease and mortality risk start to increase

**Table 1. Marginal median parameter estimates and 95% CrI for the constant IFR model.** $\beta$ is the rate of seroreversion and $\gamma$ denotes the IFR. The estimated median time to seroreversion given by $1/\beta$ is 176 (95% CrI: 159–197 days).

| Parameter | Median (95% CrI) |
| --- | --- |
| $\beta$ | 0.0057 (0.0051–0.0063) |
| $\gamma_{London}$ | 0.0049 (0.0046–0.0063) |
| $\gamma_{NorthWest}$ | 0.0080 (0.0073–0.0087) |
| $\gamma_{NortEast}$ | 0.0103 (0.0095–0.0112) |
| $\gamma_{SouthWest}$ | 0.0094 (0.0087–0.0101) |
| $\gamma_{SouthEast}$ | 0.0118 (0.0109–0.0129) |
| $\gamma_{Midlands}$ | 0.0085 (0.0079–0.0091) |
| $\gamma_{East}$ | 0.0083 (0.0077–0.0090) |

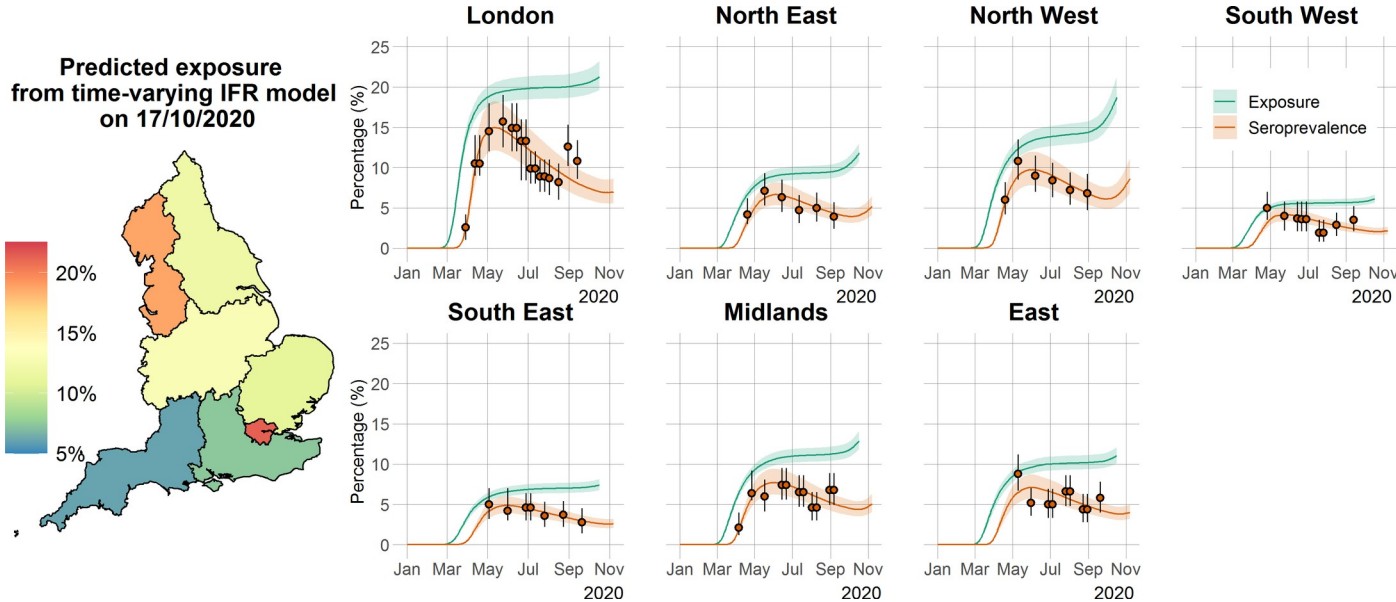

**Fig 3. Time course of the SARS-CoV-2 pandemic up to 7 November 2020 for the seven regions in England in the time-varying IFR model.** The orange solid circles and black error bars in each regional panel represent the observed seroprevalence data and their credible intervals after adjusting for the sensitivity and specificity of the antibody test. The green and orange lines show the median time-varying IFR model predictions for exposure and seroprevalence, respectively, while the shaded areas correspond to the 95% CrI. The regional median predicted exposure levels (expressed as the proportion of the population that has been infected) as of 17 October 2020 are shown on the map of England. The map base layer was obtained from: https://opendata.arcgis.com/datasets/ 8d3a9e6e7bd445e2bdcc26cdf007eac7_4.geojson.

significantly) and the estimated IFR. Interestingly, not only is the population in London younger but there is also a lower proportion of the population comprising elderly people living in care homes, which may explain the proportionally lower contribution of care-home deaths to the overall mortality in London. This covariate appears to explain more than 75% of the variance observed in estimated IFRs across regions (S4B Fig). Note that other mortality risk factors, such as diabetes and pulmonary and liver disease, seem to have no correlation with estimated IFR at all (S6 Table).

An alternative formulation of our modelling approach allows IFR to vary over time according to the stage of epidemic progression, i.e., allowing for IFR to potentially decrease as the population gains immunity, shielding of vulnerable people is optimized and patient treatment is improved. Unfortunately, it is extremely difficult to extrapolate the underlying risk of infection (a proxy for epidemic progression) from reported case data due to the volatility in testing capacity. Hence, we propose that the optimal metric for epidemic progression is the cumulative test positivity ratio. In the absence of severe sampling biases, the test positivity ratio is a good indicator of changes in underlying population infection risk, as a larger proportion of people will test positive if infection prevalence increases. In fact, it is clear from S5D Fig that the test positivity ratio is a much better indicator of exposure than the case fatality ratio (CFR) or the hospitalization fatality ratio (HFR), as it mirrors the shape of the mortality incidence curve (S5B Fig). For the time-varying IFR, we took the normalized cumulative test positivity ratio time-series and applied it as a scalar of the maximum IFR value estimated for each region (more details can be found in the Materials and Methods section).

The results from the time-varying IFR model are consistent with the results from the constant IFR model and in very good agreement with serological data (Fig 3). The mean seroreversion rate in this model was estimated to be 162 days (95% CrI: 148–186 days), a 5.5–6.9%

difference compared with the constant IFR model, meaning that the estimation for the seroreversion rate was robust to the assumption of the shape of the IFR. Critically, predictions for cumulative exposure in the population are very robust to the assumption of the shape of IFR, with both models forecasting the same levels of overall exposure (comparing Figs 2 and 3). Estimates for the time-varying IFR model (S3 Table and S6 and S7 Figs) suggest a slight decrease in IFR from March to November 2020 in several regions of England; this was most significant in London.

The methodology for estimating the cumulative exposure in the population proposed in this paper rests on several key assumptions that might be violated in practice, the first of which is the assumption of fixed delays in the infection time history as depicted in Fig 1. This is admittedly a simplification of delay-distribution approaches used elsewhere but, as determined by our sensitivity analysis, has no significant implications for the estimates of IFR and seroreversion rates for either the constant IFR model or the time-varying IFR model (S8 and S9 Figs). Indeed, relaxing the fixed delays assumption would only result in a shift in predicted exposure during the very early stages of the pandemic. Second, our methodology relies on mortality data to infer the true shape of the cumulative exposure curve. The official English government data dashboard provides two mortality datasets: 'Deaths with COVID-19 on the death certificate' and 'Deaths within 28 days of positive test' by date of death [35]. We opted for the former as the default source of mortality data used, because the latter dataset significantly underestimates COVID-19-associated deaths at the beginning of the pandemic, at a time when transmission was very high and PCR testing capacity was at its lowest (S10 Fig). Note that once testing capacity reached the tens of thousands of tests per day, the two mortality data streams report essentially the same figures. If we use 'Deaths within 28 days of positive test by date of death' as a model input, we obtain a seroreversion rate that is 8.8–12.8% shorter, along with 16–32% lower regional IFRs (S11 and S12 Figs). This is an expected consequence of having the model fit to the same serology data, while assuming there were 17,000 fewer deaths during the spring 2020 epidemic wave. More importantly, the cumulative exposure predictions are extremely robust to the explored mortality inputs (S13 and S14 Figs).

## Discussion

Given the current polarization of opinion around COVID-19 natural immunity, we realize that our results are likely to be interpreted in one of two conflicting ways: (1) the rate of seroreversion is high, therefore achieving population (herd) immunity is unrealistic, or (2) exposure in more affected geographical areas, such as London, is much higher than previously thought, and population immunity has almost been reached, which explains the decrease in IFR over time. We would like to dispel both interpretations and stress that our results do not directly support either. Regarding (1), it is important to note that the rate of decline in neutralizing antibodies, reflective of the effective immunity of the individual, is not the same as the rate of decline in seroprevalence. Antibodies may visibly decline in individuals yet remain above the detection threshold for antibody testing [6]. Conversely, if the threshold antibody titer above which a person is considered immune is greater than the diagnostic test detection limit, individuals might test positive when in fact they are not effectively immune. The relationship between the presence and magnitude of antibodies (and therefore seropositive status) and protective immunity is still unclear, with antibodies that provide functional immunity only now being discovered [13]. Furthermore, T cell-mediated immunity is detectable in seronegative individuals and is associated with protection against disease [8]. Therefore, the immunity profile for COVID-19 goes beyond the presence of a detectable humoral response. We believe our methodology to estimate total exposure levels in England offers valuable insights and a solid

evaluation metric to inform future health policies (including vaccination) that aim to disrupt transmission. With respect to (2), we must clarify that decreasing IFR trends can result from a combination of population immunity, improvement in patient treatment, better shielding of those at highest risk, and selection processes operating at the intersection of individual frailty and population age structure. We can eliminate exposure levels as the main driver of this process as there is no clear temporal signal for IFR for regions other than London. This is confirmed by data on age-dependent mortality rates at different stages of the epidemic, which show that mortality rates in London have decreased substantially since the first spring wave, much more so than in other regions (S15 Fig). As no significant disparities in treatment outcomes across regions were found [34], an alternative interpretation of IFR trends in England is that individuals who are more likely to die from infection (due to some underlying illness, being in a care home, being over a certain age or any other risk factors) will do so earlier. This means that as the epidemic progresses, selection (through infection) for a decrease in average population frailty (a measure of death likelihood once infected) is taking place and, consequently, a reduction in the ratio of deaths to infections. The lower estimated IFR for London can be attributed to the city's relatively younger population and lower rates of elderly persons in care homes when compared with populations in the other regions of England (S4 Fig, top right panel), indicating that if this selection process does exist it will be more pronounced in younger populations with a smaller subset of very frail individuals.

We should mention some details that potentially limit the applicability of the methodology presented here to other countries, especially low- and middle-income countries (LMICs). The most pertinent detail is one of data quality. Whereas our assumption that COVID-19 deaths are nearly perfectly reported in England is a plausible one, this is very unlikely to hold for other countries across the globe [36]. To account for potential under-reporting, we could include a constant or time-varying reporting ratio to transform reported deaths into 'likely' deaths. The direct consequence of using predicted deaths as a model input would be that any IFR estimates would be very difficult to disentangle from the underlying reporting ratio. The quality of the seroprevalence data itself is paramount, and the data collection protocol can have a major influence on the obtained estimates, as evidenced by two concurrent seroprevalence studies conducted in Manaus, Brazil. Whereas one study reports a raw seroprevalence of approximately 40% [37] using the Abbot test, the other reports an antibody positivity of approximately 13% [38] using the WONDFO SARS-CoV-2 antibody test, with both measured in May 2020. A significant difference is that the former study used blood donor samples, whereas the latter relates to household surveys. Another issue that is likely to be relevant to many LMICs is that the provision of a reliable level of uncertainty around the seroreversion estimate relies on having several sequential seroprevalence measurements. In countries with a very limited capacity for conducting serosurveys, we suggest using the posterior distribution for seroreversion provided here as an informative prior and proceed to estimate infection fatality ratios and total exposure profiles. However, we should note that the thresholds of seropositive and seronegative assignment vary across assays, hindering the applicability of estimates resulting from data generated with a specific assay to other settings where different assays might be used.

In conclusion, we propose a new method to forecast the total exposure to SARS-CoV-2 from seroprevalence data that accounts for seroreversion and uses daily mortality and test positivity ratio data to aid inference. The associated estimate of time to seroreversion of 176 days (95% CrI: 159–197 days) lies within realistic limits derived from independent sources [13,14,23,29,30]. The total exposure in regions of England estimated using this method is more than double the latest seroprevalence measurements. Implications for the impact of vaccination and other future interventions depend on the, as yet uncharacterized, relationships

between exposure to the virus, seroprevalence, and population immunity. To assess vaccination population impact, one can consider the population at risk to be those individuals who are seronegative, those with no past exposure (confirmed or predicted), or those with no T-cell reactivity. Here, we offer an extra dimension to the evidence base for immediate decision-making, as well as anticipating future information from the immunological research community about the relationship between SARS-CoV-2 exposure and immunity.

## Materials and methods

### Data sources

We used publicly available epidemiological data to infer the underlying exposure to SARS-CoV-2 over time, as described below:

**Regional daily deaths.** The observed daily mortality data for each of seven English regions (London, North West, North East (contains both the North East and Yorkshire and the Humber regions), South East, South West, Midlands (East and West Midlands combined) and East of England), from January 1 2020 to November 11 2020, relate to daily deaths with COVID-19 on the death certificate by date of death. This information was extracted from the UK government's official COVID-9 online dashboard [35] on March 8, 2021. The age dependent regional death rate data used to compare spring and winter 2020 waves was extracted from the same source.

**Regional adjusted seroprevalence.** Region-specific SARS-CoV-2 antibody seroprevalence measurements, adjusted for the sensitivity and specificity (82.5% and 99.1%, respectively) of the Euroimmun antibody test, were retrieved from the national COVID-19 surveillance reports produced by Public Health England [16].

**Regional test-positivity ratios.** Time series of region-specific PCR test-positivity ratios were downloaded from the UK government's official dashboard [35] on May 16, 2021

**Regional population age structure and non-COVID epidemiological indices.** Region-specific population structures were obtained from the UK Office for National Statistics 2018 population survey [39]. Other demographic and epidemiological indicators such as number of care home beds and incidence of diabetes, e.g., were extracted from the PHE online database [40], using the search terms: "care home"; "diabetes"; "pulmonary disease"; "heart disease".

### Mechanistic model

We developed a mechanistic mathematical model that relates reported daily deaths from COVID-19 to seropositive status by assuming all COVID-19 deaths are reported and estimating an IFR that is congruent with the observed seroprevalence data. For each region, $i = 1,\ldots,7$, corresponding to London, North West, North East, South East, South West, Midlands and East of England respectively; we denote the IFR at time $t$ by $\alpha_i(t)$ and the number of daily deaths by $m_i(t)$. While we formulate the model in terms of a general, time-dependent IFR, we assume its default shape to be time invariant and later allow IFR to vary with the stage of the epidemic.

Using the diagram in Fig 1 as a reference and given a number of observed deaths at time $t$, $m_i(t)$, we can expect a number of infections $\frac{1}{\alpha_i(t-d_\epsilon)} m_i(t - d_\epsilon)$ to have occurred $d_\epsilon$ days before. Of these infected individuals, $m_i(t)$ will eventually die, while the remaining $\frac{1-\alpha_i(t)}{\alpha_i(t)} m_i(t)$ will seroconvert from seronegative to seropositive. This assumes that seroconversion occurs, on average, with the same delay from the moment of infection as death.

Assuming that seropositive individuals convert to seronegative (serorevert) at a rate $\beta$, the rate of change in the number of seropositive individuals in region $i$, $X_i(t)$ is given by:

$$\frac{dX_i(t)}{dt} = \frac{1 - \alpha_i(t)}{\alpha_i(t)} m_i(t) - \beta X_i(t) \tag{1}$$

Solving Eq (1), subject to the initial condition $X_i(t_0) = 0$, where t is time since January 1, 2020, gives:

$$X_i(t) = e^{-\beta t} \int_{t_0}^{t} e^{\beta w} \frac{(1 - \alpha_i(w))}{\alpha_i(w))} m_i(w) dw \tag{2}$$

Discretizing Eq (2) with daily intervals ($\Delta w = 1$) gives:

$$X_i(t) = e^{-\beta t} \sum_{w=t_0}^{t} \left[ \frac{(1 - \alpha_i(w))}{\alpha_i(w)} e^{\beta w} m_i(w) \right] \tag{3}$$

The model-predicted proportion of seropositive individuals in each population, $x_i(t)$, is calculated by dividing $X_i(t)$ (Eq (3)) by the respective region population size at time $t$, $P_i - \sum_{w=t_0}^{t} m_i(w)$, where $P_i$ is the reported population in region $i$ before the COVID-19 outbreak [39]:

$$x_i(t) = e^{-\beta t} [P_i - \sum_{w=t_0}^{t} m_i(w)]^{-1} \sum_{w=t_0}^{t} \left[ \frac{(1 - \alpha_i(w))}{\alpha_i(w)} e^{\beta w} m_i(w) \right] \tag{4}$$

This is relatively straightforward when the serology data are already adjusted for test sensitivity and specificity, as is the case with the datasets used here. For unadjusted antibody test results, the proportion of the population that would test positive given the specificity ($k_{sp}$) and sensitivity ($k_{se}$) can be calculated as:

$$z_i(t) = k_{se} x_i(t) + (1 - k_{sp})(1 - x_i(t)).$$

As mentioned earlier, the method that we present here allows for the IFR, $\alpha_i(t)$, to be (a) constant or (b) vary over time with the stage of the epidemic:

a.  For a constant IFR, we have:

$$\alpha_i(t) = \gamma_i$$

b.  For a time-varying IFR, we first define the epidemic stage, $ES(t)$, as the normalized cumulative positivity ratio:

$$ES_i(t) = \frac{\left( \sum_{w=t_0}^{t} y_i(w - \delta_p) \right)}{\left( \sum_{w=t_0}^{T} y_i(w - \delta_p) \right)} \tag{5}$$

where $y_i(t)$ is the confirmed case positivity ratio at time $t$ in the proportion of individuals testing positive for the virus, $\delta_p$ is the average time between testing positive and seroconversion (see Fig 1) and $T$ is the total number of days from $t_0$ until the last date of positivity data. In this work, we fixed $\delta_p = 7$ days (see Fig 1 and the main text). We assume that the IFR is a linear function of the normalized cumulative positivity ratio as follows:

$$\alpha_i(t) = \gamma_i(1 - \eta_i ES_i(t)) \tag{6}$$

where $\eta_i \in [0,1]$ and $\gamma_i \in [0,1]$ are coefficients to be estimated. At the start of the epidemic,

when the epidemic stage is 0 (see Eq (5)), then $\alpha_i(t) = \gamma_i$, whereas when the epidemic stage is 1, $\alpha_i(t) = \gamma_i - \eta_i \times \gamma_i \leq \gamma_i$.

In Eq (5), $y_i(t)$ is taken from the daily regional positivity ratios provided in the UK government's data dashboard [35].

Once the model is parameterized, we can estimate the total proportion of the population that has been exposed, $E_i$, using the following formula:

$$E_i(t - \delta_\epsilon) = [P_i - \sum_{w=t_0}^{t} m(w)]^{-1} \sum_{w=t_0}^{t} \frac{1 - \alpha_i(w)}{\alpha_i(w)} m_i(w) \qquad (7)$$

where $\delta_\epsilon$ is fixed to 21 days (Fig 1).

## Observation model for statistical estimation of model parameters

We developed a Bayesian model to estimate the model parameters $\theta$ and present the posterior predictive distribution of the seroprevalence (Eq (4)) and exposure (Eq (7)) over time. The results are presented as the median of the posterior with the associated 95% credible intervals (CrI). We assumed a negative binomial distribution [41] for the observed number of seropositive individuals in region $i$ over time, $X_i^{obs}(t)$:

$$X_i^{obs}(t) = x_i^{obs}(t) \times (P_i - \sum_{w=t_0}^{t} m_i(w)) \qquad (8)$$

where $x_i^{obs}(t)$ is the observed seroprevalence in region $i$ over time. Then, the observational model is specified for region $i$ with observations at times $t_{i1}, t_{i2} \ldots, t_{in_i}$:

$$X_i^{obs}(t) \sim NB(X_i(t), \phi), t = t_{i1}, t_{i2} \ldots, t_{in_i} \qquad (9)$$

where $NB(X_i(t), \phi)$ is a negative binomial distribution, with mean $X_i(t)$–given by Eq (3)–and $\phi$ is an overdispersion parameter. We set $\phi$ to 100 to capture additional uncertainty in data points that would not be captured with a Poisson or binomial distribution. We assume uninformative beta priors for each of the parameters, according to the assumption made for how the IFR is allowed to vary over time:

a. For a constant IFR, we have $\theta = \{\{\gamma_i\}_{i=1}^{i=7}, \beta\}$ and take priors:

$$\gamma_i \sim Beta(1, 1), \beta \sim Beta(1, 1) \qquad (10)$$

b. For a time-varying IFR, we have $\theta = \{\{\gamma_i\}_{i=1}^{i=7}, \{\eta_i\}_{i=1}^{i=7}, \beta\}$ and take priors:

$$\gamma_i \sim Beta(1, 1), \eta_i \sim Beta(1, 1), \beta \sim Beta(1, 1) \qquad (11)$$

We use Bayesian inference (Hamiltonian Monte Carlo algorithm) in RStan [42] to fit the model to seroprevalence data by running four chains of 20,000 iterations each (burn-in of 10,000). We use 2.5% and 97.5% percentiles from the resulting posterior distributions for 95% CrI for the parameters. The Gelman–Rubin diagnostics ($\hat{R}$) given in S1 and S2 Tables show values of 1, indicating that there is no evidence of non-convergence for either model formulation. Furthermore, the effective sample sizes ($n_{eff}$) in S1 and S2 Tables are all more than 10,000, meaning that there are many samples in the posterior that can be considered independent draws.

## Sensitivity analyses

The results in the main text explore two model formulations: one that assumes IFR is constant over time and another that relaxes that assumption. These models share several underlying assumptions, particularly relating to time delays between events in the life history of infection, prior distributions, and data sources. To ascertain the robustness of our main results, we estimated the relevant parameters using a series of different models as listed in S4 Table. Essentially, we explore how our estimates change as we:

- assume different values for the delay between testing PCR positive and death, $\delta_p$, and for the delay between infection and death, $\delta_\epsilon$.

- use a different prior distribution for seroreversion rate.

- use a different set of mortality data. These are sourced from the same official database [35] but obey different criteria. The main results were generated using a dataset of death certificates with COVID-19 named as the cause of death, but we also apply our method to the 'Deaths within 28 days of a positive test' dataset.

The parameter estimates for the different models considered are summarized in S5 Table. Note that parameter $\delta_p$ does not appear in Eq (1) in the Materials and Methods section, thus, estimates using the constant IFR model are only sensitive to changes in $\delta_\epsilon$ (S8 Fig). Interestingly, the time-varying IFR model is relatively insensitive to $\delta_p$ (S16 Fig) since changes in $\delta_p$ have a limited impact on the shape of the Epidemic Stage (ES) curve and consequently IFR over time (S17 Fig).

## Relationship between demographic and epidemiological factors and estimated regional IFRs

Our estimates for regional IFRs were noticeably lower for London and higher for the North East, South East and South West. Since treatment outcomes are identical across regions [34], we explored which demographic and epidemiological factors could help interpret our results (S6 Table). Our objective here was not to build the most accurate predictive regression model (as this is beyond the scope of this paper), but rather to explore a multitude of covariates which might display a statistically significant correlation with the obtained IFR trends. We thus built several linear regression models with a single covariate using regional estimates (Model 2 in S4 Table) for IFR as the dependent variable, which are summarized in S6 Table. The independent variables explored were:

- Proportion of the population over a certain age breakpoint (40 to 75 years of age in 5-year intervals). We only show the results for the two most significant age breakpoints, 45 and 60.

- Deaths in the community relative to deaths in care homes.

- Care home beds per 100 people over 75 years of age.

- Diabetes prevalence.

- Chronic liver disease mortality rate (per 100,000).

- Chronic obstructive pulmonary disease mortality rate (per 100,000).

## Supporting information

**S1 Fig. Marginal posterior distributions for parameters in the constant IFR model.** The vertical lines show the median distributions, and the grey shaded regions show the 95% CrI. (TIFF)

**S2 Fig. Probability of seropositivity persistence after seroconversion.** The green curve show the probability curve from [26] and the orange curve gives the median probability curve for Models 1, 2 and 3 in our study within the corresponding 95% credible intervals defined by the shaded area. See S4 Table for details on each model's assumptions.
(TIFF)

**S3 Fig. Cumulative deaths in the seven regions of England.**
(TIFF)

**S4 Fig. Relationship between demographic and epidemiological factors and estimated regional IFRs.**
(TIFF)

**S5 Fig. Relevant epidemiological metrics in England over the course of the pandemic.** (A) Daily COVID-19 cases and tests in England from Feb 5th 2020 to Nov 7th, 2020, alongside the testing effort corrected case curve. Case correction was done by taking the number of daily tests done on May 1st and extrapolating the number of daily cases that would be reported if the testing effort had been constant over time, i.e., how many daily cases would be reported if 20,000 tests has been done every day. (B) Comparison of testing effort corrected case incidence (blue), test positivity ratio (yellow) and daily deaths per 2 million people (purple). (C) Daily reported incidence of cases, deaths and people tested up to July 1st, 2020. Note the different scale for mortality data used on panels (B) and (C). In panel (C) we present the absolute number of deaths reported per day as a means of comparing its scale to the reported case data. In panel (B) we modify the mortality incidence scale to more easily compared its shape over time against that of the daily corrected cases and test positivity ratio curves. (D) Normalized case fatality ratio (CFR), hospital fatality ratio (HFR) and PCR test positivity ratio (yellow, blue, and green lines, respectively). We assumed fixed time lags of $\delta_p$ = 14 days between PCR testing and death and $\delta_h$ = 12 days between PCR testing and hospitalization.
(TIFF)

**S6 Fig. Marginal posterior distributions for parameters in the time-varying IFR model.** The vertical lines show the median distributions, and the grey shaded regions show the 95% CrI.
(TIFF)

**S7 Fig. Posterior predictive distribution of the time-varying IFR.** The solid lines show the medians and the shaded regions show the 95% CrI.
(TIFF)

**S8 Fig. Comparison of the time course of the SARS-CoV-2 pandemic up to 7 November 2020 for the seven regions in England for the constant IFR model, given $\delta_p$ as 2 weeks and $\delta_\epsilon$ as 2, 3 and 4 weeks.** The orange solid circles and black error bars in each regional panel represent the observed seroprevalence data and their credible intervals after adjusting for the sensitivity and specificity of the antibody test. The green, red and purple lines show the median constant IFR model predictions for exposure assuming $\delta_\epsilon$ as 2, 3 and 4 weeks, respectively, while the shaded regions correspond to the 95% CrI. The green lines show the median constant IFR model predictions for seroprevalence while the shaded regions correspond to the 95% CrI. See S4 Table for details on each model's assumptions.
(TIFF)

**S9 Fig. Comparison of time course of the SARS-CoV-2 pandemic up to 7 November 2020 for the seven regions in England for the time-varying IFR model, given $\delta_p$ as 2 weeks and**

$\delta_\epsilon$ **as 2, 3 and 4 weeks.** The orange solid circles and black error bars in each regional panel represent the observed seroprevalence data and their credible intervals after adjusting for the sensitivity and specificity of the antibody test. The lines in red, green, and blue tones show the median constant IFR model predictions for exposure assuming $\delta_\epsilon$ as 2, 3 and 4 weeks, respectively, while the shaded regions correspond to their 95% CrI. The purple lines show the median constant IFR model predictions for seroprevalence while the shaded regions correspond to the 95% CrI. See S4 Table for details on each model's assumptions.
(TIFF)

**S10 Fig. Daily deaths with COVID-19 on the death certificate ("certificate death") and deaths within 28 days of positive test by date of death ("28 days positive death").**
(TIFF)

**S11 Fig. Comparison of marginal posterior distributions for estimated parameters in the constant IFR model.** The red regions show the posterior distributions for parameters using deaths within 28 days of positive test as model inputs while the blue regions show the posterior distributions of parameters using death certificate data as model inputs. See S4 Table for details on each model's assumptions.
(TIFF)

**S12 Fig. Comparison of marginal posterior distributions for estimated parameters in the time varying IFR model.** The red regions show the posterior distributions for parameters using deaths within 28 days of positive test as model inputs while the blue regions show the posterior distributions of parameters using death certificate data as model inputs. See S4 Table for details on each model's assumptions.
(TIFF)

**S13 Fig. Comparison of the time course of the SARS-CoV-2 pandemic up to 7 November 2020 for the seven regions in England for the constant IFR model between using death within 28 days of a positive COVID-19 test and death certificate data as model inputs.** The solid orange circles and black error bars in each regional panel represent the observed seroprevalence data and their credible intervals after adjusting for the sensitivity and specificity of the antibody test. The green and pink lines show the median constant IFR model predictions for exposure using death within 28 days of a positive test and death certificate data as model inputs, respectively, while the shaded regions correspond to the 95% CrIs. The purple and orange lines show the median constant IFR model predictions for seroprevalence using death within 28 days of a positive and death certificate data as model inputs, respectively, while the shaded regions correspond to the 95% CrIs. See S4 Table for details on each model's assumptions.
(TIFF)

**S14 Fig. Comparison of the time course of the SARS-CoV-2 pandemic up to 7 November 2020 for the seven regions in England for the time-varying IFR model between using death within 28 days of a positive COVID-19 test and death certificate data as model inputs.** The solid orange circles and black error bars in each regional panel represent the observed seroprevalence data and their credible intervals after adjusting for the sensitivity and specificity of the antibody test. The green and pink lines show the median constant IFR model predictions for exposure using death within 28 days of a positive test and death certificate data as model inputs, respectively, while the shaded regions correspond to the 95% CrIs. The purple and orange lines show the median constant IFR model predictions for seroprevalence using death within 28 days of a positive and death certificate data as model inputs, respectively, while the

shaded regions correspond to the 95% CrIs. See S4 Table for details on each model's assumptions.
(TIFF)

**S15 Fig. Ratio between relative rates of deaths of people who died within 28 days of their first positive test (per 100,000 population) in the winter wave of 2020 vs. the 2020 spring wave.** A ratio greater than 1 means that the age specific rate of death was greater in the winter wave than in the preceding spring wave, and vice-versa.
(TIFF)

**S16 Fig. Comparison of parameter posterior distributions for the time-varying IFR model using $\delta_p$ as 7 days (Models 4, 5 and 6), 14 days (Models 7 and 8) and 21 days (Model 9).** See S4 Table for details on each model's assumptions.
(TIFF)

**S17 Fig. Comparison of IFR estimates for seven regions in England for time-varying IFR model using $\delta_p$ as 7 days (Models 4, 5 and 6; see S4 Table for definitions of the different Models), 14 days (Models 7 and 8) and 21 days (Model 9).** See S4 Table for details on each model's assumptions.
(TIFF)

**S1 Table. The effective sample size ($n_{eff}$) and the Gelman--Rubin diagnostic ($\hat{R}$) for the eight model parameters in the default model (constant infection fatality ratio, IFR).**
(DOCX)

**S2 Table. The effective sample size ($n_{eff}$) and the Gelman--Rubin diagnostic ($\hat{R}$) for the 15 model parameters in the time-varying IFR model.**
(DOCX)

**S3 Table. Marginal median parameter estimates and 95% CrI for the time-varying IFR model.**
(DOCX)

**S4 Table. Summary of sensitivity analyses performed for $\delta_p$, $\delta_c$, death inputs and $\beta$ prior for both constant IFR and time varying IFR models.** Figs 2 and 3 of the main text were generated using Models 2 and 7 respectively.
(DOCX)

**S5 Table. Summary of parameter estimates (median and 95% credible intervals) for all models explored (as defined in S4 Table).**
(DOCX)

**S6 Table Linear regression models exploring relationships between demographic and epidemiological factors and estimated regional IFRs Each row refers to a unique linear regression model and indicates which covariate was used, alongside the resulting slope and intercept estimates (with accompanying 95% CIs) and p-value.**
(DOCX)

## Acknowledgments

We would like to thank Adam Bodley for scientific writing assistance (according to Good Publication Practice guidelines) and editorial support.

## Author Contributions

**Conceptualization:** Lisa J. White, Ricardo Aguas.

**Data curation:** Siyu Chen.

**Formal analysis:** Siyu Chen, Jennifer A. Flegg.

**Methodology:** Siyu Chen, Jennifer A. Flegg.

**Supervision:** Lisa J. White, Ricardo Aguas.

**Writing – original draft:** Lisa J. White, Ricardo Aguas.

**Writing – review & editing:** Siyu Chen, Jennifer A. Flegg, Lisa J. White, Ricardo Aguas.

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
