## [Decision Letter · Decision Letter 0]

11 May 2021

Dear Dr Aguas,

Thank you very much for submitting your manuscript "Levels of SARS-CoV-2 population exposure are considerably higher than suggested by seroprevalence surveys" for consideration at PLOS Computational Biology.

As with all papers reviewed by the journal, your manuscript was reviewed by members of the editorial board and by several independent reviewers. In light of the reviews (below this email), we would like to invite the resubmission of a significantly-revised version that takes into account the reviewers' comments.

This manuscript presents an original methodological approach to correct for seroreversion in the context of seroprevalence surveys for Covid-19. Reviewers 1 and 3 identify important issues requiring the attention of the authors. Among them I highlight the importance to address:

i) the need to discuss the sensitivity of the proposed approach to key assumptions; simple simulations under different scenarios could add to the assessment of the robustness of the proposed correction methodology.

ii) the need to adjust the topics in the discussion section to the relevant material actually dealt with in the manuscript.

iii) the need to add references supporting the claims invoked along with the modeling exercise.

In addition to the above, the reviewers' remaining issues should also receive careful attention from the authors.

We cannot make any decision about publication until we have seen the revised manuscript and your response to the reviewers' comments. Your revised manuscript is also likely to be sent to reviewers for further evaluation.

Sincerely,

Claudio José Struchiner, M.D., Sc.D.

Associate Editor

PLOS Computational Biology

Nina Fefferman

Deputy Editor

PLOS Computational Biology

This manuscript presents an original methodological approach to correct for seroreversion in the context of seroprevalence surveys for Covid-19. Reviewers 1 and 3 identify important issues requiring the attention of the authors. Among them I highlight the importance to address:

i) the need to discuss the sensitivity of the proposed approach to key assumptions; simple simulations under different scenarios could add to the assessment of the robustness of the proposed correction methodology.

ii) the need to adjust the topics in the discussion section to the relevant material actually dealt with in the manuscript.

iii) the need to add references supporting the claims invoked along with the modeling exercise.

In addition to the above, the reviewers' remaining issues should also receive careful attention from the authors.

Reviewer's Responses to Questions

**Comments to the Authors:**

Reviewer #1: Chen et al. present a clear and simple model to link together three key metrics for evaluating the progress of an epidemic, applied to the context of SARS-CoV-2 in England: antibody seropositivity, infection incidence and number of deaths. The authors use data on these three metrics to estimate the antibody seroreversion rate and region-specific infection fatality ratios. In doing so, the cumulative number of infections in England are estimated, showing that cross-sectional seroprevalence data underestimate the true extent of the SARS-CoV-2 epidemic in England to date. Estimates for the IgG (spike) seroreversion rate and IFR are broadly consistent with other studies, which supports the validity of these findings.

This is a really nicely written, concise and well conducted study. It does have a number of methodological similarities to the cited Imperial College report on a similar topic, but in my opinion, this is a nice pared-down version that provides some new and interesting insights. I feel the analyses and most of the text are already very polished, so none of my comments are particularly critical. I would just suggest a few clarifications, additional discussion points and optional sensitivity analyses around the fixed parameter assumptions.

The shared code looks good and is well documented, thanks!

##### Major comments #####

The discussion section is very brief and should include limitations and discuss sensitivity to key assumptions (namely the reliance on death data as absolute truth, though see below). For example, for the English data, the IFR will likely change depending on whether “deaths within 28 days of a positive test” or “deaths with COVID-19 on the death certificate” are used. More specific comments are in the “Minor comments” below, but in general the authors should discuss the study limitations.

The other discussion-based comment I have is on how well age structure can explain the disparate IFR estimates. The authors place a lot of weight on a younger average age explaining the lower IFR in London. This makes sense, but the paper offers no formal analyses to explore the point. Are the other region-level differences in IFR also consistent with differences in age structure? For example, the estimate for the South East is quite a lot higher than the East. Another way to think about it is that previous studies on IFR are often stratified by age group – would combining region-specific age distributions with a single age stratified IFR curve give mean IFRs that are consistent with these estimates, or would there still be unexplained residual differences? I would encourage the authors to think about how they might be able to support their hypothesis quantitatively, either through additional analyses or a more thorough consideration of demographic data. At the very least they should soften the language around “age structure explains these differences” without faithfully discussing caveats and alternative hypotheses.

##### Minor comments #####

There are three key fixed assumptions in this model which could be relaxed or explored with sensitivity analyses:

1. First, the assumption that reported tests represent exactly 100% of true COVID-19-associated deaths. I think this is a fine assumption, but there may be some violations eg. very early in the epidemic not all deaths were tested. To make this framework generalizable to other settings where death reporting and testing is not quite so rigorous as England/the UK, would it be possible to add an additional scaling parameter (possibly time-varying) to the model representing death reporting? This could then be fixed or estimated with an informative prior (eg. the posterior for IFR and beta might include 95% prior probability that death data represents 90%-110% of true deaths). To clarify, this isn’t a change that is necessary for publication, but it may help apply this model to other settings. Another consideration (which would be interesting in the publication) might be to re-fit the model using the “Deaths within 28 days of a positive test” data from the same source to get an idea of how much this affects the estimates. I don’t think you need to get into the politically charged territory of “what is a COVID-19 death?”, but it might have implications for the seroreversion rate.

2. Second, the assumption that reporting delays are fixed and constant over time. There are a few parts to this point. One is that the cited studies on the symptoms-to-death and symptoms-to-discharge delays are from China in the early stages of the pandemic. There might be reason to suspect that these delays differ both by region (based on how patients are treated etc) and over time (based on burden on healthcare services). In fact, the assumption of a 14 day value for delta_p does not fully square with the citations. Furthermore, a 2-day delay between onset and PCR test is quite short and is likely an underestimate for periods of high testing demand. These are quite pedantic points because I very much doubt changing the delays by a few days will affect the results, but these are strong assumptions which are likely to be violated in reality. I would suggest discussing the limitations of these. Furthermore, the choice of a fixed delay is a simplification of the delay-distribution approaches used elsewhere, which could also be acknowledged (ie. integrating a gamma distribution of delays rather than using a single value). A way to approach these concerns could be re-running the model under a few different assumptions for the delays to get an idea of whether or not the estimates are affected.

3. Third, averaging test positivity rates between Pillar 1 and Pillar 2. This feels like quite a crude approach, given how much test positivity can reflect changing coverage in the different test populations as well as the state of the epidemic. Again, I suspect this won’t affect the results too much, but I do wonder if things might change if Pillar 1 OR 2 were used. Another option might be to use PCR prevalence data from REACT or ONS, which albeit not incidence either, will be unbiased by changing testing capacity and coverage.

I am not sure if there are any studies on age-specific antibody waning rates, but did the authors consider including location-specific seroreversion rate parameters? Given the argument for location-specific IFRs being driven by different age distributions, the same might be true of seroreversion rates. If identifiability is a concern, then the authors could use a multi-level model (ie. beta_i ~ Normal(beta_mean, var_beta))

Further, the seroreversion rate is found to be higher here than in the Imperial report. Could the authors speculate on why this is?

Please clarify the antigen target of the assays referred to (eg. Euroimmun® -> spike).

Was there much correlation between the posteriors for IFRs and the seroreversion rate? If so, this would be very useful to report (possibly with 2-dimesional posterior density plots).

I may have missed something, but I really cannot see how eta and gamma can be estimated separately based on Equation 6. They are both just constants bounded between 0 and 1. Are the equations in this section correct, or have I just missed something obvious? Should it be alpha_i(t) = gamma_i*(1-eta_i * ES_i(t))? As with the seroreversion rate, if these parameters are correlated that would be useful to show.

##### Line-specific comments #####

L150: I am not sure I follow that seroreversion on average 6 months post seroconversion *explains* a 0.17 odds ratio of reinfection within 6 months. It is certainly a related finding, but is the implication a) that reinfections did occur because of rapid seroreversion or b) reinfections were rare because seroreversion mostly happens after 6 months (and isn’t the referenced study 5 months)? This could be made clearer.

L155: I don’t think the relevance of serological data in a broad sense is called into question. Certainly, how these data are interpreted should be, but the data are still incredibly useful!

Figure 2: this is a lovely figure, but I do wonder if the data from Fig S3A should be included here. Those data are part of the fitting procedure, so it is a little misleading to show the fits just to seroprevalence. For example, how does the model estimate an uptick in exposures/seroprevalence in November in the North East and North West? This seems like magic based on the seroprevalence data alone. Furthermore, what was the rationale behind showing the map as of 17th October or just arbitrary (changes to the tier system)?

L162: I hate to be that person, but these are credible intervals, not confidence intervals.

L217: I love this tactful start to the discussion.

L290: It might be clearer to use (t-d_e) for the infections here.

L309: Please give the specificity and sensitivity of the assay.

Eq1: Should it be beta * X_i (t)?

L335: I’m not sure how this is a hierarchical model – you just present the likelihood and priors. If you had gamma_i ~ Normal(gamma, sigma) then that would make sense, but all of the parameters are independent.

L345: was there any rationale behind setting phi to 100, or just arbitrary?

L354: No action points, just that you drew an impressive number of effective samples!!!

L356: Rogue line break.

Reviewer #2: The study provides estimates of seroreversion, IFR and population exposure levels.

The results are nicely presented. The methods seem sound and well-justified for this reviewer. It is difficult to place the results in terms of known reports of cases and deaths (and case fatality ratio). It would be beneficial to discuss the connection of these results to case and death data (and CFR) for clear interpretation.

Reviewer #3: Suggested Plos Comp. review points (as presented in https://journals.plos.org/ploscompbiol/s/reviewer-guidelines#loc-criteria-for-publication):

1. What are the main claims of the paper and how significant are they for the discipline?

The work brings up the challenge to truly estimate the proportion of the population exposed to SARS-COV-2, the severity of the disease determined by the IFR and the seroreversion time, a limit marker for detecting antibodies in an infected individual. The authors estimate seroreversion time around six months, infection fatality rate varying from 0.0049 to 0.0118, and variations of the population exposed to SARS-COV-2 from 5% to 20% in seven regions in England as of October 2020. Their mathematical modelling relied on data of observed daily mortality, SARS-CoV-2 antibody seroprevalence measurements, and weekly positivity ratios of laboratory-confirmed COVID-19 cases aggregated for each of the regions studied. The results obtained can significantly help in the pandemic and guide the application of non-pharmaceutical and vaccinations strategies. However, the authors change the focus of the work objective in the Result and Discussion sections and bring up a hypothetical analysis and discussions about the frailty selection process. Their method and data do not represent biological, social or economic factors, which influence the concept of frailty, which is instead an unobserved susceptibility to adverse outcomes. Thus, the paper must be not accepted for publication unless major corrections are performed.

2. Are these claims novel? If not, which published articles weaken the claims of originality of this one?

The methodology and results concerning the estimation of the population exposure, fatality rate and time of seroreversion are very similar to the work of Shioda, Kayoko, et al. "Estimating the cumulative incidence of SARS-CoV-2 infection and the infection fatality ratio in light of waning antibodies." medRxiv (2020), https://doi.org/10.1101/2020.11.13.20231266, posted on November 16, 2020. Despite this, the analysis performed by the draft submitted by Aguas still has originality and is adapted to a different setting.

3. Are the claims properly placed in the context of the previous literature? Have the authors treated the literature fairly?

No. In some parts of the text, the authors lack literature review to support their claims. Please, see the General Comments below.

4. Do the data and analyses fully support the claims? If not, what other evidence is required?

Please, see the General Comments below.

5. Would additional work improve the paper? How much better would the paper be if this work were performed and how difficult would it be to do this work?

The work would significantly improve if additional (not complicated) analysis would be performed and the text rewritten to address the primary objective proposed by the authors. Please, see the General Comments below.

6. Are original data deposited in appropriate repositories and accession/version numbers provided for genes, proteins, mutants, diseases, etc.?

Yes, all data and codes are available.

7. Has the author-generated code that underpins the findings been made publicly available?

Yes.

8. Are details of the methodology sufficient to allow the experiments to be reproduced?

Yes.

9. Is the manuscript well organized and written clearly enough to be accessible to non-specialists?

yes

10. Does the paper use standardized scientific nomenclature and abbreviations? If not, are these explained at the first usage?

yes

General comments

Comment 1: The authors do not mention the findings regarding the estimated fatality rate in the abstract.

Comment 2: In the manuscript, the authors mention that the “REACT” study had estimated a 13% exposure proportion in London, from June 20th to July 13th 2020. How do these values compare to estimates made by the authors for a more extended period up to November 7th? What is the estimated exposure and IFR in the literature for the other regions compared to the results obtained in Figure 2? for which period?

Comment 3: The authors claim that reducing the ratio of deaths to infections is due to frailty (lines 175-180); however, they do not consider the seasonality of respiratory diseases, which usually has a lower prevalence in hot seasons. Also, improvement of healthcare managements and patients treatments can be an issue for a reduction in IFR. The work lacks discussion about that.

Comment 4: Figure 2 and 3 are repeated in the manuscript.

Comment 5: A delicate issue in their analysis is the assumption of uninformative beta priors for the parameters estimated. Uninformative priors are vague and may affect the posterior distribution of the results. Also, in the literature, it is shown that the time from symptom onset to seroconversion is consistent with the Weibull distribution. It must be taken into account in the analysis performed.

Comment 6: The applicability of the proposed methods to other places worldwide can be influenced by the serological assays methodologies, which affect, for instance, the effects of variations on the time lag between exposure and seroconversion. How would it affect the analysis performed? How much does it influence the measures obtained in the results? What are the recommendations and limitations of the applicability of the work worldwide?

Comment 7: Additionally, the discussion section is poorly written and does not compare the work with others performed on the literature. For instance, the author mention:

“The associated estimate of time to seroreversion of 176 days (95% CrI: 159-197) lies within realistic limits derived from independent sources.” Which sources (references)?

“The total exposure in regions of England estimated using this method is more than double the last seroprevalence measurements.” In which references?

Reviewer #4: I read the manuscript with interest. The authors have designed and carried out a good study, involving COVID19 data on seroprevalence in England and mathematical modeling analysis of it.

As I understand from the authors' modeling analyses of the English seroprevalence data and the conclusion arrived at, the time-invariant IFR model (results in Figure 2) is as good as the time-dependent IFR model (results in Figure 3). This is a good news from the model parsimony point-of-view - a simple model performs as good as a slightly more complex one does. However, I do not see a connection between this ("no statistically significant time-dependence on IFR was inferred") and the authors' point on line 197 "... suggesting this phenomenon is dependent on age structure." The authors have not performed their modeling analyses using an age-structured model. Therefore, the authors need to observe some caution before arriving at what they said on the line. As I think, it needs a little bit more unpacking. Or modification in the statement.

The data presented in Figures 2 and 3 have calendar months as independent variable. But the authors describe the data in the units of weeks of 2020 (lines 270 to 273). A reader like myself starts wondering which week number corresponds to which calendar month.

The authors have made a simplifying assumption in their models, "From the moment of exposure, individuals seroconvert a fixed 21 days later ..." I was expecting to see some sensitivity analysis around this assumption of a fixed 21-day period: how will the [main] result change if this fixed time-period is shortened or lengthened by a week?

**Have the authors made all data and (if applicable) computational code underlying the findings in their manuscript fully available?**

Reviewer #1: Yes

Reviewer #2: Yes

Reviewer #3: Yes

Reviewer #4: **No: **I did not see a mention of freely available computer codes of the mathematical models developed/employed by the authors for producing the results. Ditto about the seroprevalence data.

PLOS authors have the option to publish the peer review history of their article (what does this mean?). If published, this will include your full peer review and any attached files.

Reviewer #1: **Yes: **James Hay

Reviewer #2: No

Reviewer #3: No

Reviewer #4: No
---

## [Decision Letter · Decision Letter 1]

4 Aug 2021

Dear Dr Aguas,

Thank you very much for submitting your manuscript "Levels of SARS-CoV-2 population exposure are considerably higher than suggested by seroprevalence surveys" for consideration at PLOS Computational Biology. As with all papers reviewed by the journal, your manuscript was reviewed by members of the editorial board and by several independent reviewers. The reviewers appreciated the attention to an important topic. Based on the reviews, we are likely to accept this manuscript for publication, providing that you modify the manuscript according to the review recommendations.

Two reviewers offer minor suggestions for improvement and we would like to invite the authors to fix these final details if appropriate.

Sincerely,

Claudio José Struchiner, M.D., Sc.D.

Associate Editor

PLOS Computational Biology

Nina Fefferman

Deputy Editor

PLOS Computational Biology

[LINK]

Two reviewers offer minor suggestions for improvement and we would like to invite the authors to fix these final details if appropriate.

Reviewer's Responses to Questions

**Comments to the Authors:**

Reviewer #1: My initial comments on this manuscript were very positive, recommending minor revisions to explore the impact of some key, fixed assumptions (namely the death data and reporting delays) on the estimated seroreversion rate and IFR. The authors have responded very thoroughly and added some clear sensitivity analyses to address these concerns. The arguments against some of my suggestions are convincing (e.g., not including a death reporting rate parameter). I also think the extended discussion describing the frailty hypothesis more clearly is a great addition. Overall, I think the authors have done a fantastic job revising the manuscript and recommend publication.

##### Minor comments #####

-I would mention in the supplementary figure captions showing the different scenarios that the scenario assumptions are shown in Table S4. Same for eg. the legend in Figure S11. I would just try to make it clearer which numbers refer to which assumptions, and be consistent with terms like “Model 2” vs. “Scenario 2”.

-Regarding the regression analysis explaining region-specific differences in IFRs: there seems to be a slight discrepancy between the response document and Figure S4. The response document states that % of people over 45 explains 61% of the variance, whereas the figure shows proportion over 60 as the cutoff. The main text also refers to the proportion of people in care homes explaining 75% of variance, not the figure for people over 45.

-Related to the new regression analyses: the description of what was done is too brief, and the regression coefficients and CIs should be reported. It is also not quite clear if the authors fit a single regression including all of the demographic covariates, or if they fit separate simple linear regressions to each. I actually think I could be missing a file: the authors refer to supplementary methods, but I cannot find them anywhere in the system. Please just check these are included and that the regression methods/results are described in detail. Same goes for the delta_p/delta_e analyses.

-Response 1.21: the methods still refer to this as a hierarchical model (L427).

-L38 in author summary: typo “take thin into account”.

-Inconsistent use of “” and ‘’ eg. L127/128.

Reviewer #2: The authors have sufficiently considered my comments. The manuscript is improved considering all reviewer comments.

Reviewer #3: no

Reviewer #4: I congratulate the authors for revising their original manuscript as per the comments/suggestions made by all reviewers. However, I realized that the authors might have forgotten to run spelling checker or a final proof-reading - some words need to be corrected. For example, in line 38 thin -> this?

In addition, I see that the authors still use the word "hierarchical" in line 427 of the revised submission. They had responded to Comment 1.21 saying that [and I quote then here, "We agree that this is not technically a hierarchical model and have changed the text to reflect that."] that they would drop this word. So, what happened? I ask authors to be mindful of following things in their revision as per their own responses.

I do not have any further comment.

**Have the authors made all data and (if applicable) computational code underlying the findings in their manuscript fully available?**

Reviewer #1: Yes

Reviewer #2: Yes

Reviewer #3: Yes

Reviewer #4: Yes

PLOS authors have the option to publish the peer review history of their article (what does this mean?). If published, this will include your full peer review and any attached files.

Reviewer #1: **Yes: **James A Hay

Reviewer #2: No

Reviewer #3: No

Reviewer #4: No

Figure Files:

Data Requirements:

Reproducibility:

References:

---

## [Editor Report · Decision Letter 2]

8 Sep 2021

Dear Dr Aguas,

We are pleased to inform you that your manuscript 'Levels of SARS-CoV-2 population exposure are considerably higher than suggested by seroprevalence surveys' has been provisionally accepted for publication in PLOS Computational Biology.

Best regards,

Claudio José Struchiner, M.D., Sc.D.

Associate Editor

PLOS Computational Biology

Nina Fefferman

Deputy Editor

PLOS Computational Biology

---

## [Editor Report · Acceptance letter]

16 Sep 2021

PCOMPBIOL-D-21-00584R2 

Levels of SARS-CoV-2 population exposure are considerably higher than suggested by seroprevalence surveys

Dear Dr Aguas,

I am pleased to inform you that your manuscript has been formally accepted for publication in PLOS Computational Biology. Your manuscript is now with our production department and you will be notified of the publication date in due course.

With kind regards,

Andrea Szabo
